# A High-Yield Process for Production of Biosugars and Hesperidin from Mandarin Peel Wastes

**DOI:** 10.3390/molecules25184286

**Published:** 2020-09-18

**Authors:** Eun Jin Cho, Yoon Gyo Lee, Jihye Chang, Hyeun-Jong Bae

**Affiliations:** 1Bio-energy Research Center, Chonnam National University, Gwangju 61186, Korea; choej47@gmail.com; 2Department of Bioenergy science and Technology, Chonnam National University, Gwangju 61186, Korea; spake123@naver.com (Y.G.L.); 825wlgp@naver.com (J.C.)

**Keywords:** mandarin peel waste, biomass waste, hesperidin, biosugar, value-added products

## Abstract

In this research, novel biorefinery processes for obtaining value-added chemicals such as biosugar and hesperidin from mandarin peel waste (MPW) are described. Herein, three different treatment methods were comparatively evaluated to obtain high yields of biosugar and hesperidin from MPW. Each method was determined by changes in the order of three processing steps, i.e., oil removal, hesperidin extraction, and enzymatic hydrolysis. The order of the three steps was found to have a significant influence on the production yields. Biosugar and hesperidin production yields were highest with method II, where the processing steps were performed in the following order: oil removal, enzymatic hydrolysis, and hesperidin extraction. The maximum yields obtained with method II were 34.46 g of biosugar and 6.48 g of hesperidin per initial 100 g of dry MPW. Therefore, the methods shown herein are useful for the production of hesperidin and biosugar from MPW. Furthermore, the utilization of MPWs as sources of valuable materials may be of considerable economic benefits and has become increasingly attractive.

## 1. Introduction

Environmental pollution is among the most serious problems faced by humanity today [1]. The United Nations Food and Agriculture Organization (FAO) has estimated that approximately 1.3 billion tons of food is wasted worldwide every year, with fruits and vegetables having the highest wastage rates among all the food types [2]. A third of all the produced fruits and vegetables is wasted even before reaching the consumer due to programmed overproduction and unfulfillment of retailer quality standards [3,4]. Most of the fruit waste is currently dumped or disposed in landfills worldwide, which leads to significant environmental problems and increased financial cost [5]. Therefore, efforts are concentrated on reducing the amount of fruit waste. Recycling or reusing of fruit waste can effectively reduce the amount of waste sent to landfills and incinerators, and thereby also environmental pollution [6]. Therefore, there is an urgent need to develop processes for production of value-added chemicals from recycled fruit wastes.

Mandarin (*Citrus unshiu*) is a citrus plant with a very popular fruit, whose production continues to increase consistently throughout the world [7]. Worldwide mandarin production reported to FAO Corporate Statistical Database (FAOSTAT) in 2018 was 53.6 million tons, including tangerines, clementines, and satsumas. The primary processing waste of mandarin fruit, for juice production and other food material, is the peels of the fruit, which are usually discarded. After juice extraction, this mandarin peel waste (MPW) accounts for approximately 27% of the wet fruit mass [8]. The annual amount of MPW produced worldwide is estimated to be 14.5 million tons. The MPW is disposed of through a burning process or used as raw material for active substance extraction and animal feed manufacturing. However, these means are not satisfactory as the market demand may be limited or secondary contamination and low-value products the result. Several application areas have been proposed for MPW, such as pectin, flavonoid, fiber, and animal feed production [9,10,11]. However, a large amount of this waste is still dumped annually and needs to be managed and industrialized properly.

Mandarin is as a rich source of flavonoid, especially flavone glycosides. Hesperidin is the predominant flavone glycoside in mandarin. Hesperidin structure comprises hesperetin and disaccharide rutinose. Hesperidin exhibits anti-oxidant, anti-inflammatory, anti-cancer, anti-diabetic, anti-allergic, UV-protective, analgesic, radical-scavenging, and hypolipidemic properties [12,13,14,15,16]. Hesperidin was also recently proposed as a drug substance with promising activity against coronavirus disease (COVID-19), which has evolved into a global pandemic. From this viewpoint, hesperidin showed remarkable binding affinity to the three main protein receptors of severe acute respiratory syndrome coronavirus 2 (SARS-CoV-2), i.e., the cause of COVID-19, and may thus inhibit the proteins responsible for viral development and infection [17,18,19,20,21]. As a result of these functions, hesperidin is used extensively in the pharmaceutical and food industries, and its demand is increasing worldwide [22,23]. The global hesperidin market was valued at USD 81 million in 2019 and is expected to reach USD 125.2 million by the end of 2026, growing at a Compound Annual Growth Rate (CAGR) of 6.3% during the period 2021–2026 [24]. Therefore, it is required to seek an attractive resource for large-scale production of hesperidin. The peel includes the highest concentrations of flavonoids such as hesperidin in the mandarin fruit [25]. This high concentration and the large amount of peel waste available render MPW a rich source of naturally occurring flavonoids.

The development of an effective and selective method for bioactive compound extraction would be beneficial. Recent extraction methods have largely focused on finding a green method that minimizes the use of solvent and energy, and reduces wastes, such as subcritical water extraction, ultrasound extraction, microwave extraction, supercritical fluids extraction, and pressurized hot water extraction (PHWE) [26,27,28,29,30,31,32]. More recently, the enzyme-assisted extraction method has shown faster extraction, higher recovery, reduced solvent usage and lower energy consumption when compared to non-enzymatic methods [33,34].

Biosugars have attracted significant attention in several fields of carbohydrate research and synthesis. Biosugars can be used as raw materials to obtain carbon sources for fermentation processes, which are used to produce high-value products [35]. However, suitable raw materials are required to achieve economically viable biosugar production processes [36]. MPW is rich in fermentable and soluble biosugars, such as glucose, fructose, and sucrose, and cellulose and hemicellulose, and thus may be used as raw material or biosugar production as well. 

Herein, we report a method to produce biosugars and hesperidin using MPW as raw material (Figure 1).

## 2. Results

### 2.1. Design of Experimental Methods

We first designed three methods for effective production of biosugar and hesperidin from MPW, as shown in Figure 2. Each method differed from each other in terms of the order of three processing steps, i.e., oil removal, hesperidin extraction, and enzymatic hydrolysis. The oil content of MPW was removed via diethyl ether extraction. Extraction of hesperidin from MPW was conducted using the methanol extraction method described previously [37], which was successfully used to improve hesperidin extraction from citrus peel. The methanol has also been proven for the extraction of phenolic compounds from citrus fruits [38]. Enzymatic hydrolysis was used to obtain biosugars from MPW. Method I involved first the use of enzymatic hydrolysis of MPW to obtain biosugars followed by oil removal and hesperidin extraction from the residue obtained after enzymatic hydrolysis. In method II, the MPW was first subjected to diethyl ether extraction to remove oil, which was followed by enzymatic hydrolysis and hesperidin extraction. Finally, method III was conducted by first performing the oil removal, then hesperidin extraction, and finally enzymatic hydrolysis.

### 2.2. Chemical Composition of MPW after the Application of Different Treatment Methods

We determined the chemical composition of the material before the enzymatic hydrolysis step of each method (i.e., after grinding, oil removal, and hesperidin extraction in methods I, II, and III, respectively). As shown in Table 1, glucose was the main carbohydrate constituent of untreated MPW (24.8%, based on dry matter). Rhamnose, arabinose, xylose, mannose and galactose were also present, albeit at low concentrations. After removal of oil (method II), the carbohydrate concentrations were slightly increased. This indicates that oil removal did not significantly modify the cellulose and hemicellulose structures. In contrast, although there was a slight increase in total carbohydrate concentrations, the glucose concentration with method III was reduced by 35% than that with method I. However, the concentration of hemicelluloses, including xylose, arabinose, mannose and galactose, was increased with a remarkable increase in arabinose concentration (by approximately 2.64-fold than that with method I). This is due to the destruction and solubilization of celluloses in methanol [39], which may result in an apparent reduction in the glucose contents in the cellulose. The increase in the arabinose concentration may also be attributed to an increase in the hemicellulose concentration due to the removal of cellulose and phenolic compounds under diethyl ether/methanol extraction conditions. We further confirmed the existence of sugars in the methanol extract using HPLC. Appendix A shows that the methanol extract contained glucose, fructose, and sucrose in small amounts.

### 2.3. Optimization of Enzyme Loading

The costs of enzyme production and raw material purchases are the two leading contributors to the overall biosugar production expenses [40,41]. Herein, two in-house produced enzymes (cellulase and pectinase) were used to increase the enzymatic conversion rate through synergistic action. Different amounts of cellulase and pectinase were added to 1% MPW (*w*/*v*) to evaluate the effects of different enzymes on the MPW hydrolysis yields. The highest conversion yield was achieved in the range of 60.0–84.8% (*w*/*w*) using pectinase (2.3–18.4 mg/g) and in the rage of 56.3–79.6% (*w*/*w*) using cellulase (3.4–27.2 mg/g), respectively. Thus, pectinase appears to be the main enzyme active in MPW degradation. Although the highest final glucose concentration was obtained from enzyme loading (27.2 mg cellulase/g MPW and 18.4 mg pectinase/g MPW), enzymatic hydrolysis of MPW was conducted using a dose of 3.4 mg of cellulase/g MPW and 9.2 mg of pectinase/g MPW for optimal enzyme loading to reduce enzyme cost (Figure 3). In order to maintain a high yield of reducing sugar with more economical value, lower enzyme dosage with significant reducing sugar yield must be chosen as the efficient hydrolysis [42]. If the enzyme loading can be reduced, the cost of enzyme will reduce naturally. These results also reveal that pectinase and cellulase acted synergistically during the hydrolysis of the MPW. When the combined of cellulase (3.4–27.2 mg/g) and pectinase (2.3–18.4 mg/g) were used, the conversion yield was improved by 81.8–92.9% (*w*/*w*) in comparison to the addition of each enzyme alone. 

### 2.4. Biosugar Production via Enzymatic Hydrolysis

Biosugar production performance of each MPW treatment method was investigated using enzymes at different combinations of concentrations (Figure 4). The synergistic effects of the enzymes on the release of total monosaccharides and glucose, xylose, galactose, arabinose, and fructose are shown in Figure 4. The combination of cellulase and pectinase increased the biosugar yields from all samples. The highest biosugar yield was obtained when cellulase and pectinase were applied together on the diethyl ether-treated MPW (method II). Sugar content of de-oiled MPW (method II) was 1.15 and 1.46 times higher than that of untreated MPW (method I) and diethyl ether/methanol treated MPW (method III), respectively. Although the amount of arabinose was higher in method III than that in other methods, the total biosugar yield was lower. This result is consistent with chemical composition analysis results. Extraction was found to increase the digestibility of MPW, possibly due to the removal of oils and non-polar compounds which inhibit enzyme degradation. To confirm the removal of oils from the MPW, the composition of diethyl ether extract was also analyzed using GC–MS (Appendix A). Appendix A shows that this extract included fatty acids, alcohols, aldehydes, hydrocarbons, sterols, tocopherol, and flavonoids. Therefore, diethyl ether effectively removed oils from the MPW.

### 2.5. Isolation and Characterization of Hesperidin Obtained MPW

Crude hesperidin obtained using each treatment methods was purified via recrystallization using DMSO [38], and its chemical structure was identified by spectroscopy. The UV and ^1^H NMR spectra of commercial and hesperidin purified herein were evaluated (Figure 5B,C). Spectra of crude and purified hesperidin were similar to each other. The UV spectra of the hesperidin showed absorption peaks at 290, 315, and 345 nm. The results from ^1^H NMR of purified hesperidin exhibited a major peak of isolated hesperidin from MPW, which is very similar to the major peak of the hesperidin standard. After purification, some peaks on the NMR spectrum disappeared. The purity of hesperidin in DMSO-d_6_ was determined at 91.2% using ^1^H NMR. 

The amounts of extracted hesperidin using each method were compared after recrystallization using DMSO. The highest amount of hesperidin was extracted using method II. In addition, the extraction of hesperidin was enhanced by enzymatic hydrolysis, regardless of whether oil removal was performed. This is evident from the 1.36-fold higher hesperidin concentration with method II than that with method III, where extraction was performed without enzymatic hydrolysis. Therefore, enzymatic hydrolysis was effective in enhancing the recovery of hesperidin from MPW.

### 2.6. Overall Mass Balance

An overall mass balance diagram was constructed for each step in the treatment process, including oil removal, methanol extraction, and enzymatic hydrolysis steps (Figure 6). The oil fraction in MPW was removed from 0.91% *w*/*w* (method I) to 1.29% *w*/*w* (method II and III) after ether extraction. The highest amounts of biosugars and hesperidin were found to be produced using method II, which enabled production of 34.46 g of total sugars and 6.48 g of hesperidin from 100 g of MPW (dry weight). As such, 3.3- and 1.36-times higher yields were obtained for biosugars and hesperidin, respectively, than those obtained using method III. The production efficiencies of the three methods were ordered in decreasing order as follow: method II > method I > method III. Moreover, enzymatic hydrolysis was also found to be effective for the extraction of hesperidin from MPW.

## 3. Materials and Methods

### 3.1. Materials

Mandarin fruits were purchased from the local market. After peeling the fruits, the peels were freeze-dried, and then ground using an electric grinding machine. Standard hesperidin was purchased from Sigma-Aldrich (Seoul, Korea). Diethyl ether, methanol, DMSO, and acetic acid were obtained from Ducksan (Seoul, Korea). All other chemicals were of at least analytical grade purity. The water used in all procedures was de-ionized and filtered using a US Filter purification system (Merck Millipore, Burlington, MA, USA).

### 3.2. Carbohydrate Analysis

The neutral sugar composition of MPW was determined using gas chromatography (GC, Shimadzu, Otsu, Japan) via injection into a DB-225 capillary column (30 m × 0.25 mm i.d., 0.25 μm film thickness, J&W; Agilent, Folsom, CA, USA) under previously reported conditions [43]. 

### 3.3. Optimization of Enzyme Loading

In-house produced cellulase and pectinase were used [44]. Cellulase and pectinase activities were determined using the National Renewable Energy Laboratory method [45]. Cellulase and pectinase were added to MPW at concentrations within a range of 3.4−27.2 mg protein/g MPW and 2.3−18.4 mg protein/g MPW, respectively, to determine the optimum enzyme-loading levels. Enzymatic hydrolysis of the substrate (1.0% *w*/*v*) was performed in 50 mM sodium citrate buffer (pH 4.8) for 24 h at 45 °C. The amounts of soluble sugars in the resulting mixture was then measured using high-performance liquid chromatography (HPLC). HPLC was performed using a refractive index detector (2414, Waters, Milford, MA, USA) equipped with a Rezex RPM column (4.6 × 300 mm; Phenomenex, CA, USA). HPLC-grade water was supplied at a flow rate of 0.6 mL/min as a mobile phase at a controlled temperature of 80 °C. Each sample was measured in triplicate. A three-dimensional contour plot and response surface were created using Sigma plot 10.0 software (Systat Software GmbH, Erkrath, Germany).

### 3.4. Enzymatic Hydrolysis

Enzymatic hydrolysis of the MPW was conducted in a 10 mL of solution containing 1.0% (*w*/*w*) dry matter and in-house produced cellulase (3.4 mg/g MPW) and pectinase (9.2 mg/g MPW) for 24 to 48 h at 45 °C. These loadings were used because they yielded the highest glucose during the previous hydrolysis steps carried out on untreated MPW for enzyme-loading optimization. The soluble sugar content was analyzed using HPLC.

### 3.5. Removal of Oil Fraction from MPW

Extraction was performed using a Soxhlet apparatus. First, 15 g of dried MPW or treated MPW was weighed into an extraction thimble (Whatman, internal length 5 mm, internal diameter 19 mm, Maidstone, UK), and extracted using 200 mL of diethyl ether at 45 °C. After 5 h of extraction, the MPW was air-dried at room temperature under a hood until the odors of organic solvents were not detected. The presence of organic solvents was confirmed again using solid-state ^13^C NMR spectroscopy. Finally, oil yield was determined. Diethyl ether was evaporated under vacuum, and the sample was subjected to GC/MS analysis.

### 3.6. Methanol Extraction of Hesperidin

The treated MPW was re-extracted using 50 mL of methanol for 3 h at 75 °C. The extracted MPW was dried and weighted. Methanol content of the extract was removed using a rotary evaporator (EYELA, Tokyo, Japan). The filtrate was then acidified using 50 mL of 5% acetic acid. The solid precipitate containing crude hesperidin was filtered out and washed with 5% acetic acid. The crude hesperidin was freeze-dried and stored at 4 °C until later use. Each experiment was repeated at least three times [38].

### 3.7. Purification of Hesperidin

Crude hesperidin was mixed with dimethyl sulfoxide, and the mixture was then heated up to 60−80 °C. Solid concentration in this mixture was 5% (*w*/*w*). Next, an equal volume of water was added slowly while stirring. The mixture was left to crystallize for a few hours. Hesperidin crystals were then filtered off and washed, first with warm water and then with isopropanol and ethanol. The resulting powder was then dried and stored in desiccators until constant weight readings were recorded [38].

### 3.8. GC/MS Analysis of Oil Fraction

The analysis of the diethyl ether extract was performed using GC/MS (Shimadzu, GCMS-QP2010 Ultra) via injection into a DB-5 column (30 m × 0.25 mm, 0.25 μm film thickness). A 1.0 mL of a solution including helium as the carrier gas at a concentration of 10 mL mL^−1^ (split ratio of 1:30) was injected at a constant flow rate of 1.2 mL min^−1^. The injector temperature was 250 °C, and the ion source temperature was 200 °C. The components of the oil were identified on the basis of comparison of their relative indices and mass spectra by computer matching with Synthetic Compounds GC/MS library from Shimadzu (Columbia, MD, USA) and National Institute of Standards and Technology (NIST08) libraries provided with the computer controlling GC/MS system [46]. Compounds of interest were considered if they had a similarity match of at least 75%. The sample obtained by method II was used for the GC/MS analysis. 

### 3.9. Characterization of Hesperidin

The NMR spectra were recorded at 400 MHz for ^1^H by Bruker NMR spectrometer (AVANCE III HD 400, Karlsruhe, Germany) using DMSO-d_6_ and chemical shifts were given on a δ (ppm) scale with tetramethylsilane (TMS) as the internal standard. A Perkin Elmer Lambda 35 UV/Vis spectrophotometer (Waltham, MA, USA) was used for UV spectrum. The solution was scanned from 200 to 800 nm. Methanol was used as the background for the hesperidin sample.

## 4. Conclusions

In summary, we have developed an integrated methodology, incorporating oil removal, enzymatic hydrolysis, and hesperidin extraction to enable the co-production of biosugar and hesperidin from MPW. Enzymatic treatment was effective in enhancing the extraction yield of hesperidin from MPW. Additionally, the results revealed that the combination of enzymes had a significant increasing effect on the biosugar yield in comparison with using each enzyme separately. Among the methods tested here, the method involving first oil removal followed by hesperidin extraction and enzymatic hydrolysis yields the highest biosugar and hesperidin production efficiency. The methodology shown herein may open a new avenue for utilization of the abundant citrus waste biomass to produce biosugars and hesperidin.

## Figures and Tables

**Figure 1 molecules-25-04286-f001:**
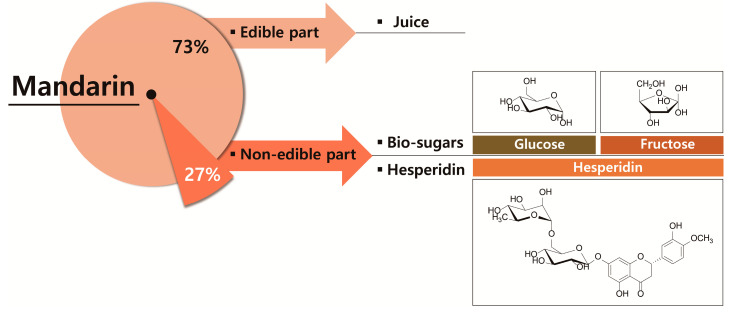
Process for production of valuable products from mandarin peel waste (MPW).

**Figure 2 molecules-25-04286-f002:**
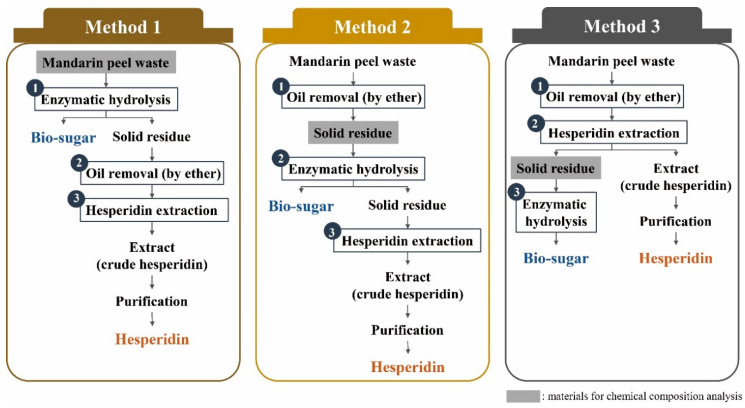
Schematic of the three methods used to produce biosugars and hesperidin from mandarin peel waste (MPW).

**Figure 3 molecules-25-04286-f003:**
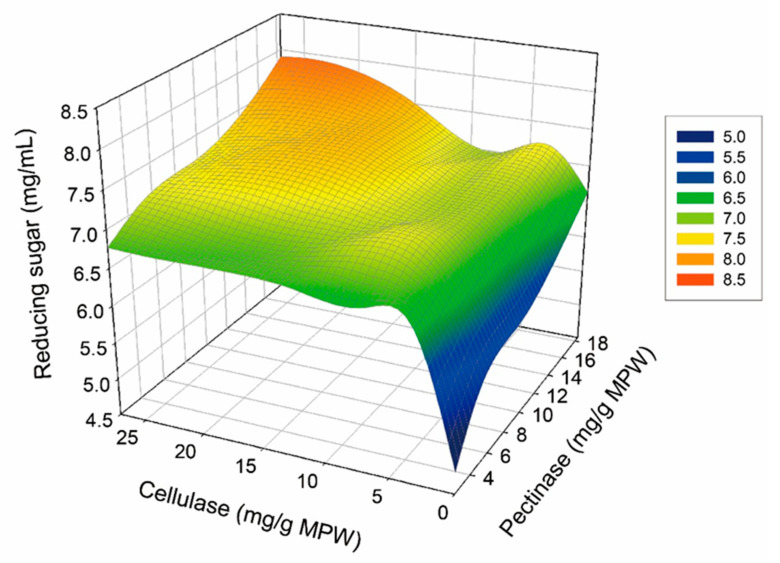
Response surface profile of reducing sugar concentration (mg/mL) under various of enzyme-loading levels.

**Figure 4 molecules-25-04286-f004:**
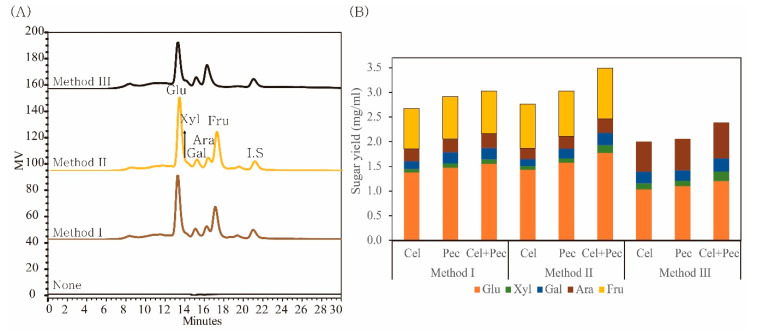
(**A**) HPLC spectra sugar yields obtained using different methods, and (**B**) the synergistic effect of cellulase and pectinase on MPW with each method. Glu: glucose, Xyl: xylose, Ara: arabinose, Fru: fructose, Gal: galactose, I.S.: internal standard, Cel: cellulase, Pec: pectinase.

**Figure 5 molecules-25-04286-f005:**
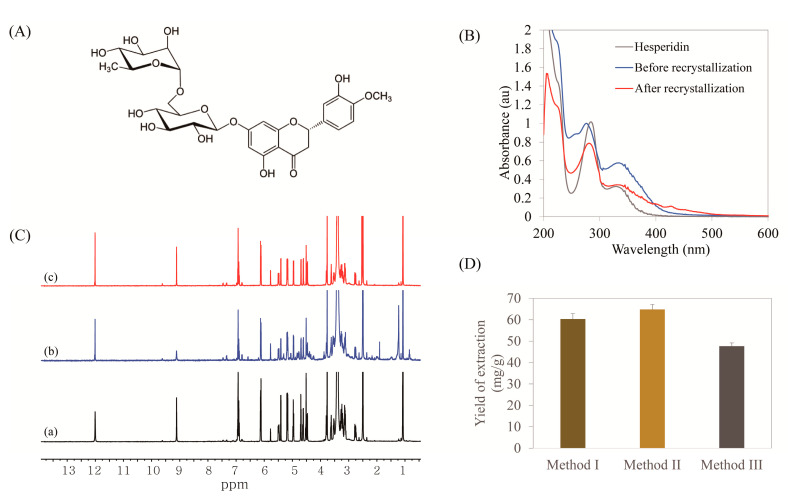
(**A**) Structure of hesperidin. (**B**) UV–vis spectra of commercial hesperidin and hesperidin before and after purification. (**C**) ^1^H NMR spectra of (a) commercial hesperidin, (b) hesperidin before purification, and (c) after purification. (**D**) The yield of hesperidin extraction with each method.

**Figure 6 molecules-25-04286-f006:**
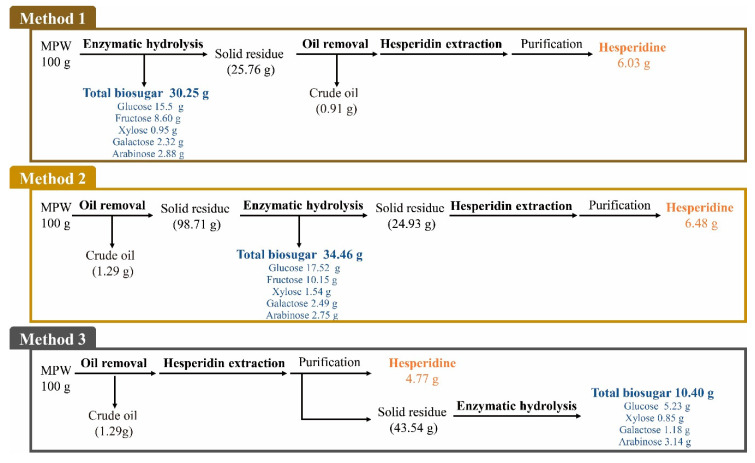
Overall mass balance for each method to produce biosugar and hesperidin from MPW.

**Table 1 molecules-25-04286-t001:** Monosaccharide composition (% dry weight) of mandarin peel waste (MPW) before the enzymatic hydrolysis step of each method.

(%)	Method I	Method II	Method III
Rhamnose	2.7 ± 0.1	2.9 ± 0.5	2.0 ± 0.1
Arabinose	3.6 ± 0.0	3.6 ± 0.1	9.5 ± 1.2
Xylose	1.4 ± 0.1	1.4 ± 0.0	2.8 ± 0.3
Mannose	1.2 ± 0.1	1.2 ± 0.0	1.7 ± 0.1
Galactose	1.8 ± 0.0	1.9 ± 0.1	4.1 ± 0.4
Glucose	24.8 ± 1.0	25.8 ± 0.7	16.1 ± 1.0
Total	35.5 ± 1.3	36.8 ± 1.4	36.1 ± 2.7

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
