# Peer review of "A High-Yield Process for Production of Biosugars and Hesperidin from Mandarin Peel Wastes"

_molecules, 2020, doi:10.3390/molecules25184286_

Round 1
Reviewer 1 Report
Review of molecules-919769
Line 43-44 Where did you get this statistic from? Can you please provide a reference?
Line 86-87 Can you please provide the reference to the National Renewable Energy Laboratory method used?
Line 106-107 Are you saying that you smelled the sample to ensure that the organic solvent had completely evaporated? This is a safety concern.
Line 108-109 How was the diethyl ether evaporated?
Lines 180-182 Can you provide any references on the cost of producing in-house enzymes compared to purchasing commercial enzymes?
Line 186 Please review this sentence. I think it should read, “although 27.2 mg pectinase /g MPW and 18.4 mg/g MPW cellulase yielded the highest final glucose 186 concentration (Figure 3).”
Line 187 The authors state there was a synergistic effect. But the amount of sugars obtained from the use of the combination of pectinase and cellulase are not much greater than each enzyme alone. Can the authors prove that the effective is not just additive?
Figure 4. Can the quality of the text in these images be improved?
Figure 4. If abbreviations are to be used in the figures they must be denoted in the figure caption.
Figure 5. Can the quality of the nmr spectra be improved?
Figure S2 This figure is included in the supplementary material but is not mentioned in the text of the manuscript. Can you please make reference to it at the appropriate place in the text?
Author Response
Manuscript ID: Molecules-919769 Title: A High-Yield Process for Production of Biosugars and Hesperidin from Mandarin Peel Wastes Thank you so much for your useful comments and suggestions of our manuscript. We have modified the manuscript accordingly, and detailed corrections are listed below point by point. In revised manuscript, the order of “Materials and methods” section and “Results” section has changed to fit the journal’s form. Line 43-44 Where did you get this statistic from? Can you please provide a reference? → The value is calculated based in FAOSTAT statistics and reference 8. Line 86-87 Can you please provide the reference to the National Renewable Energy Laboratory method used? → We added the reference. Line 106-107 Are you saying that you smelled the sample to ensure that the organic solvent had completely evaporated? This is a safety concern. → We performed the experiment very carefully. However, this could be a safety problem, so we have revised this part for clarity. Line 108-109 How was the diethyl ether evaporated? →The diethyl ether solvent has evaporated using a rotary evaporator under vacuum at 35℃. This information was added in the paragraph 3.5 (before paragraph 2.5). Lines 180-182 Can you provide any references on the cost of producing in-house enzymes compared to purchasing commercial enzymes? → No paper has yet been published on the cost of producing in-house enzymes compared to purchasing commercial enzymes. Zhuang et al. reported the computer simulation data for large scale of enzyme production processes. The results indicated that the cellulase enzyme production are $15.67 per kilogram ($/kg)and $40.36/kg, for the solid state cultivation (SSC) and traditional submerged fermentation (SmF) methods, respectively, while the market price for the cellulase enzyme is $90.00/kg (in 2004) [1]. [1] Zhuang, Jun & Marchant, Mary A. & Nokes, Sue & Strobel, Herbert, 2007. Economic Analysis of Cellulase Production Methods for Bio-Ethanol. Biosystmes and Agricultural Engineeing Faculty Publications. 102. Line 186 Please review this sentence. I think it should read, “although 27.2 mg pectinase/g MPW and 18.4 mg/g MPW cellulase yielded the highest final glucose 186 concentration (Figure 3).” → We have rechecked and revised this sentence. And the figure 3 has been corrected to match the text. Line 187 The authors state there was a synergistic effect. But the amount of sugars obtained from the use of the combination of pectinase and cellulase are not much greater than each enzyme alone. Can the authors prove that the effective is not just additive? → The combination of pectinase and celluase did not explain the strong synergistic interaction. However, the conversion yield was improved by 81.8- 92.9% (w/w) in comparison to the addition of each enzyme alone, when the combined cellulase (3.4-27.2 mg/g) and pectinase (2.3-18.4 mg/g) are used. The highest conversion yield was achieved in the range of 60.0-84.8% (w/w) using pectinase and in the rage of 56.3-79.6 % (w/w) using cellulase, respectively. The paragraphs have been completely rewritten. Figure 4. Can the quality of the text in these images be improved? → The figure 4 has been changed as requested. Figure 4. If abbreviations are to be used in the figures they must be denoted in the figure caption. → We added the abbreviations in the figure caption. Figure 5. Can the quality of the nmr spectra be improved? → We improved the NMR spectra in the figure 5. Figure S2 This figure is included in the supplementary material but is not mentioned in the text of the manuscript. Can you please make reference to it at the appropriate place in the text? → We mentioned it in the paragraph 2.4 (before paragraph 3.4).Reviewer 2 Report
The work entitled “A High-Yield Process for Production of Biosugars and Hesperidin from Mandarin Peel Wastes” reports a protocol for the valorisation of Mandarin wastes, recovering high added value compounds (i.e. hesperidin, biosugars and oily fraction). The authors provided an interesting investigation, focused on the influence of treatment step order, describing yield variations of the target compounds. A structured characterization approach has been adopted.
The paper is well written and generally consistent in the experimental and discussion section, providing an interesting waste valorisation protocol. Anyhow, it is opinion of this referee that, before the publication, some minor revisions are required.
- In particular, soundness should be given to the paper background. Some implementations into the introduction section are suggested, as it follows:
Due to the paper's hot topic, i.e. waste valorisation, some space should be given to green chemistry/green extraction approach, providing even more consistency to the work background. This referee suggest the following references:
- Tang, S.; Bourne, R.; Smith, R.; Poliakoff, M. The 24 Principles of Green Engineering and Green Chemistry: “IMPROVEMENTS PRODUCTIVELY”. Green Chem. 2008, 10, 268–269.
- Chemat, F.; Abert Vian, M.; Cravotto, G. Green Extraction of Natural Products: Concept and Principles. J. Mol. Sci. 2012, 13, 8615–8627.
Considering the huge amount of citrus waste produced worldwide, a brief overlook on general residues exploitation (as mentioned in pag 2 line 45) should be added. This referee suggest, as an example, the following references:
- Sharma, K.; Mahato, N.; Cho, M.H.; Lee, Y.R. Converting citrus wastes into value added products: Economical and environment friendly approaches, Nutrition, 2017, 34, 29-46.
- Fidalgo, A.; Ciriminna, R.; Carnaroglio, D.; Tamburino, A.; Cravotto, G.; Grillo, G.; Ilharco, L.; Pagliaro, M.; Eco-Friendly Extraction of Pectin and Essential Oils from Orange and Lemon Peels, ACS Sustainable Chem. Eng., 2016, 4, 2243–2251.
- Gomez-Mejia, E.; Rosales-Conrado, N.; Leon-Gonzalez, M.E.; Madrid, Y. Citrus peels waste as a source of value-added compounds: Extraction and quantification of bioactive polyphenols. Food Chem, 2019, 295, 289-299.
Furthermore, a short section should be added focusing on hesperidin recovery/extraction from citrus, providing the relative specific literature. This is fundamental to better contextualize the work background.
- Beside the introduction implementations, some minor revision are suggested as it follows:
Paragraph 2.3. ”Optimization of enzyme loading“: The authors should report which program was used for the response surface profiling and the relative parameters. Furthermore, reference of the applied NREL method should be provided.
Paragraph 2.6 and 2.7. “Methanol extraction of hesperidin”, “Purification of hesperidin”: Both the paragraphs should contain reference of the selected extraction and purification method, as reported in Results and Discussion.
Paragraph 2.8. “GC/MS analysis of oil fraction”: The paragraph should report the library used to identify the compound reported in SI (Table S1). In addition, percentual matching quality of library with the analytes should be reported (average, in Material and Methods, or for every molecules, in the SI).
Results and Discussion: Oil fraction yields were cited in the experimental section (pag 3 line 107-8), but not in the discussion. The subject was not addressed further. Moreover, some considerations should be added if considerable composition differences were detected or not, between the three extraction method (1, 2 or 3). Concerning to that, the composition and the chromatogram reported in the SI (Table S1 and Figure S2) do not report to which extraction method they belong.
Paragraph 3.2. “Chemical composition of the obtained material following the application of different treatment methods on MPW“: For sake of clarity, the title should be reduced, synthetizing the concept
Pag 5 line 185-6: It is opinion of the referee that, for sake of comparison, reducing sugar yields of the cited enzymes concentrations should be reported in the text. In this way, Figure 3 could be better understood. Furthermore, a brief statement should be added concerning the decision of the authors to do not choose the maximum yield of sugars.
Figure 5 A: It is not clear what do the “A, B, C” letters mean.
Figure 5 B: It is opinion of this referee that absorbance of the reference should be evidenced with a more visible colour (i.e. red). Thereby, the comparison would be easier. Furthermore, lines thickness should be slightly increased.
Figure 5 C: The H-NMR spectrum should be cut near after the 0 and the 12, gaining some space to enlarge the figure. Furthermore, the caption reports UV-vis spectra instead of H-NMR.
Figure 6: Concerning the mass balance, the yield of oil removal should be reported. The arrow for enzymatic hydrolysis in method 3 it is not visible enough.
- Minor spell check:
Pag 1 line 14: It is opinion of this referee that this sentence should be revised, for sake of clarity.
Pag 2 line 74: “ground” should be corrected with the past tense.
Pag 3 line 99: “the most glucose”.
Pag 4 line 50: “usefulness methanol”.
Pag 6 line 213-4: the sentence concerning NMR should be rephrased.
Pag 8 line 249: typo in “efficiency” spelling.
Author Response
Manuscript ID: Molecules-919769 Title: A High-Yield Process for Production of Biosugars and Hesperidin from Mandarin Peel Wastes Thank you so much for your useful comments and suggestions of our manuscript. We have modified the manuscript accordingly, and detailed corrections are listed below point by point. In revised manuscript, the order of “Materials and methods” section and “Results” section has changed to fit the journal’s form. • In particular, soundness should be given to the paper background. Some implementations into the introduction section are suggested, as it follows: Due to the paper's hot topic, i.e.waste valorisation, some space should be given to green chemistry/green extraction approach, providing even more consistency to the work background. This referee suggest the following references: - Tang, S.; Bourne, R.; Smith, R.; Poliakoff, M. The 24 Principles of Green Engineering and Green Chemistry: “IMPROVEMENTS PRODUCTIVELY”. Green Chem. 2008, 10, 268–269. - Chemat, F.; Abert Vian, M.; Cravotto, G. Green Extraction of Natural Products: Concept and Principles. J. Mol. Sci. 2012, 13, 8615–8627. → Referring to your opinion, we added the more background about green extraction approach in the introduction section. Considering the huge amount of citrus waste produced worldwide, a brief overlook on general residues exploitation (as mentioned in pag 2 line 45) should be added. This referee suggest, as an example, the following references: - Sharma, K.; Mahato, N.; Cho, M.H.; Lee, Y.R. Converting citrus wastes into value added products: Economical and environment friendly approaches, Nutrition, 2017, 34, 29-46. - Fidalgo, A.; Ciriminna, R.; Carnaroglio, D.; Tamburino, A.; Cravotto, G.; Grillo, G.; Ilharco, L.; Pagliaro, M.; Eco-Friendly Extraction of Pectin and Essential Oils from Orange and Lemon Peels, ACS Sustainable Chem. Eng., 2016, 4, 2243–2251. - Gomez-Mejia, E.; Rosales-Conrado, N.; Leon-Gonzalez, M.E.; Madrid, Y. Citrus peels waste as a source of value-added compounds: Extraction and quantification of bioactive polyphenols. Food Chem, 2019, 295, 289-299. → The information about general residues exploitation of citrus waste has been added in the revised manuscript. Furthermore, a short section should be added focusing on hesperidin recovery/extraction from citrus, providing the relative specific literature. This is fundamental to better contextualize the work background. → A new paragraph has been added in the revised manuscript as requested. • Beside the introduction implementations, some minor revision are suggested as it follows: Paragraph 2.3. ”Optimization of enzyme loading“: The authors should report which program was used for the response surface profiling and the relative parameters. Furthermore, reference of the applied NREL method should be provided. → The information was added in the paragraph 3.3 (before paragraph 2.3). We also added the reference about NREL method. Paragraph 2.6 and 2.7. “Methanol extraction of hesperidin”, “Purification of hesperidin”: Both the paragraphs should contain reference of the selected extraction and purification method, as reported in Results and Discussion. → We added the reference in both paragraph. Paragraph 2.8. “GC/MS analysis of oil fraction”: The paragraph should report the library used to identify the compound reported in SI (Table S1). In addition, percentual matching quality of library with the analytes should be reported (average, in Material and Methods, or for every molecules, in the SI). → The information was added in the paragraph 3.8 (before paragraph 2.8) Results and Discussion: Oil fraction yields were cited in the experimental section (pag 3 line 107-8), but not in the discussion. The subject was not addressed further. Moreover, some considerations should be added if considerable composition differences were detected or not, between the three extraction method (1, 2 or 3). Concerning to that, the composition and the chromatogram reported in the SI (Table S1 and Figure S2) do not report to which extraction method they belong. → Oil fraction yields were added in the paragraph 2.6 (before paragraph 3.6). GC/MS was analyzed using sample obtained by method II. This was explained in the supplementary data. Paragraph 3.2. “Chemical composition of the obtained material following the application of different treatment methods on MPW“: For sake of clarity, the title should be reduced, synthetizing the concept → The title of paragraph 3.2 was reduced to “Chemical composition of MPW after the application of different treatment methods”. Pag 5 line 185-6: It is opinion of the referee that, for sake of comparison, reducing sugar yields of the cited enzymes concentrations should be reported in the text. In this way, Figure 3 could be better understood. Furthermore, a brief statement should be added concerning the decision of the authors to do not choose the maximum yield of sugars. → The reducing sugar yields of the used enzymes concentrations was included in the paragraph 2.3 (before paragraph 3.3). And the reason for determination of enzyme loading dose has been briefly stated in the revised manuscript. Figure 5 A: It is not clear what do the “A, B, C” letters mean. → The letters were incorrectly inserted. Therefore, they were removed in the Figure 5A. Figure 5 B: It is opinion of this referee that absorbance of the reference should be evidenced with a more visible colour (i.e. red). Thereby, the comparison would be easier. Furthermore, lines thickness should be slightly increased. → The figure 5b has been changed as requested. Figure 5 C: The H-NMR spectrum should be cut near after the 0 and the 12, gaining some space to enlarge the figure. Furthermore, the caption reports UV-vis spectra instead of H-NMR. → We changed the 1H-NMR spectrum and a caption for figure. Figure 6: Concerning the mass balance, the yield of oil removal should be reported. The arrow for enzymatic hydrolysis in method 3 it is not visible enough. → The yield of oil removal was added in revised manuscript. And the figure 6 has been changed as per the suggestion. • Minor spell check: Pag 1 line 14: It is opinion of this referee that this sentence should be revised, for sake of clarity. → This sentence has been rewritten. Pag 2 line 74: “ground” should be corrected with the past tense. → This is the past tense of the word grind. Pag 3 line 99: “the most glucose”. → We have changed the word from “the most glucose” to “the highest glucose” Pag 4 line 50: “usefulness methanol”. → Correction has been made in the revised manuscript. Pag 6 line 213-4: the sentence concerning NMR should be rephrased. → This sentence has been changed in text. Pag 8 line 249: typo in “efficiency” spelling. → We corrected the spelling of the word.Reviewer 3 Report
The article is interesting and actual. The subject may have a direct interest for the readers and may have direct application.
This is exactly one of the issues that I would like to reinforce. The authors must improve the introduction section in order to demonstrate the importance of the article, for example, to industrial applications.
Figure 1 is not exactly the type of figure that the scientific articles have. Please, reformulate the figure to something more technical or scientific. This one is too childish for an article like this.
Conclusions must be expanded.
References list is too short as well. In a simple research in Google Scholar using the keywords presented in the article I found 9 similar studies that are not referenced in the article.
Author Response
Manuscript ID: Molecules-919769
Title: A High-Yield Process for Production of Biosugars and Hesperidin from Mandarin Peel Wastes
Thank you so much for your useful comments and suggestions of our manuscript. We have modified the manuscript accordingly, and detailed corrections are listed below point by point.
In revised manuscript, the order of “Materials and methods” section and “Results” section has changed to fit the journal’s form. The authors must improve the introduction section in order to demonstrate the importance of the article, for example, to industrial applications.
→ Referring to your opinion, we added the more information in the introduction section.
Figure 1 is not exactly the type of figure that the scientific articles have. Please, reformulate the figure to something more technical or scientific. This one is too childish for an article like this.
→ Figure 1 has been changed as requested. Conclusions must be expanded. → Conclusion section has been expanded.
References list is too short as well. In a simple research in Google Scholar using the keywords presented in the article I found 9 similar studies that are not referenced in the article.
→ We added more references in the revised manuscript.
Round 2
Reviewer 3 Report
The authors answered my questions and for that I consider that the article can be published.